# Soluble ST2 as a Useful Biomarker for Predicting Clinical Outcomes in Hospitalized COVID-19 Patients

**DOI:** 10.3390/diagnostics13020259

**Published:** 2023-01-10

**Authors:** Mikyoung Park, Mina Hur, Hanah Kim, Chae Hoon Lee, Jong Ho Lee, Hyung Woo Kim, Minjeong Nam, Seungho Lee

**Affiliations:** 1Department of Laboratory Medicine, Eunpyeong St. Mary’s Hospital, College of Medicine, The Catholic University of Korea, Seoul 03312, Republic of Korea; 2Department of Laboratory Medicine, Konkuk University School of Medicine, Seoul 05030, Republic of Korea; 3Department of Laboratory Medicine, Yeungnam University College of Medicine, Daegu 42415, Republic of Korea; 4Department of Laboratory Medicine, Korea University Anam Hospital, Seoul 02841, Republic of Korea; 5Department of Preventive Medicine, College of Medicine, Dong-A University, Busan 49201, Republic of Korea

**Keywords:** soluble ST2, biomarker, prediction, clinical outcome, COVID-19

## Abstract

Soluble suppression of tumorigenesis-2 (sST2) is an emerging biomarker for sepsis as well as for heart failure. We investigated the prognostic utility of sST2 for predicting clinical outcomes in hospitalized coronavirus disease 2019 (COVID-19) patients. In a total of 52 hospitalized COVID-19 patients, sST2 levels were measured using the ichroma ST2 assay (Boditech Med Inc., Chuncheon-si, Gang-won-do, Republic of Korea). Clinical outcomes included intensive care unit (ICU) admission, ventilator use, extracorporeal membrane oxygenation (ECMO) use, and 30-day mortality. sST2 was analyzed according to clinical outcomes. sST2, sequential organ failure assessment (SOFA) score, critical disease, and 4C mortality score were compared using the receiver operating characteristic (ROC) curve and Kaplan–Meier methods for clinical outcomes. The sST2 level differed significantly according to ICU admission, ventilator use, ECMO use, and 30-day mortality (all *p* < 0.05). On ROC curve analysis, sST2 predicted ICU admission, ventilator use, ECMO use, and 30-day mortality comparable to SOFA score but significantly better than critical disease. sST2 predicted ICU admission, ventilator use, and ECMO use significantly better than the 4C mortality score. On Kaplan–Meier survival analysis, hazard ratios (95% confidence interval) were 8.4 (2.7–26.8) for sST2, 14.8 (3.0–71.7) for SOFA score, 1.8 (0.5–6.5) for critical disease, and 11.7 (3.4–40.1) for 4C mortality score. This study demonstrated that sST2 could be a useful biomarker to predict ICU admission, ventilator use, ECMO use, and 30-day mortality in hospitalized COVID-19 patients. sST2 may be implemented as a prognostic COVID-19 biomarker in clinical practice.

## 1. Introduction

The coronavirus disease 2019 (COVID-19) pandemic has caused globally over six million deaths as of November 8, 2022, since its outbreak occurred in December 2019 [1,2]. The clinical spectrum of COVID-19 ranges from asymptomatic to critical disease [3]. Critical disease can present an acute respiratory distress syndrome (ARDS), sepsis, septic shock, thromboembolism, or multi-organ failure (MOF) [3,4,5]. MOF can occur in the lungs, heart, liver, kidney, and nervous system and may also present with hematological abnormalities [4]. The sequential organ failure assessment (SOFA) score is used as a severity index for assessing sepsis in COVID-19 [3,5]. Recently, 4C mortality score was developed as a prognostic index to predict the in-hospital mortality of COVID-19 [6]. 4C mortality score was calculated using age, sex, comorbidities, respiratory rate, peripheral oxygen saturation on room air, Glasgow Coma Scale, levels of urea, and C-reactive protein (CRP), and it predicted the in-hospital mortality better than the SOFA score [6].

When severe acute respiratory syndrome coronavirus 2 (SARS-CoV-2) enters host cells through the angiotensin-converting enzyme 2 (ACE2) receptor, it releases inflammatory cytokines, including interleukin (IL)-1, IL-6, or tissue necrosis factor-alpha (TNF-α) [4,5,7,8]. Overproduced cytokines induce the cytokine storm leading to MOF or death [4,7,8]. COVID-19 can cause cardiovascular disease (CVD), including heart failure (HF), and it can lead to sudden cardiac death [3,4,9,10]. COVID-19 patients with HF have a greater risk of poor outcomes in COVID-19 [3,9,10,11]. In COVID-19, MOF, ICU admission, and in-hospital mortality were significantly higher in the HF group than in the non-HF group [11]. Cardiac biomarkers could play an important role in predicting the prognosis of COVID-19 [12]. Traditional biomarkers such as natriuretic peptides and troponins were associated with COVID-19 disease severity and mortality [10,12,13].

Soluble suppression of tumorigenesis-2 (sST2), the soluble isoform of ST2, a member of the IL-1 superfamily, is an emerging prognostic biomarker for sepsis as well as for HF [12,13,14,15,16,17]. sST2 is not only released by vascular congestion but also by inflammatory and pro-fibrotic stimuli [14]. Previous studies suggested that sST2 might be a prognostic biomarker to predict the 30-day and in-hospital mortality of sepsis [14,15,16,17]. When IL-33 binds to sST2, IL-33/sST2 inhibits the release of anti-inflammatory cytokines, followed by the activation and release of inflammatory cytokines like IL-6 and TNF-α, leading to inflammation [13,14,15,16,17,18]. In addition, previous studies have reported that sST2 would predict COVID-19 disease severity and prognosis [18,19,20,21,22,23,24,25,26,27,28,29,30]. The sST2 level increased according to COVID-19 disease severity [23,25]. sST2 was associated with intensive care unit (ICU) admission, ventilator use, thrombosis, and mortality in COVID-19 [22,24,25,26,27,28,29,30].

Our previous studies showed that the biomarker-based approach would be useful for predicting clinical outcomes in COVID-19 compared with a clinical index including SOFA score and COVID-19 disease severity (critical disease) [5,31]. To the best of our knowledge, few studies have evaluated the prognostic performance of sST2 compared with clinical assessments in COVID-19 [29]. We aimed to explore the prognostic utility of sST2 for predicting clinical outcomes in hospitalized COVID-19 patients compared with clinical assessments such as SOFA score, critical disease, and 4C mortality score. In addition, we explored sST2 levels according to the 4C mortality score.

## 2. Materials and Methods

### 2.1. Study Population

The enrollment and clinical outcomes of the study population are presented in Figure 1. From February to May 2020, a total of 396 COVID-19 patients at the Yeungnam University Medical Center (YUMC), Daegu, Republic of Korea, were studied. We excluded 255 patients who visited the outpatient clinic without routine blood tests in clinical practice or who were younger than 20 years. From 141 hospitalized patients over 20 years of age, we further excluded 89 patients without available residual ethylene-diamine-tetraacetic acid (EDTA) plasma samples after routine blood tests and known 30-day status. Finally, 52 patients were enrolled. Among the 52 patients, 44 patients were included in a previous study [5]. The study population had no specific limitations on care at enrollment. The symptom duration ranged up to 25 days. Therefore, we could not enroll the study population at the same point in the COVID-19 disease course. In a total of 52 patients, a follow-up period for 30-day status was five months from February to the end of June 2020. No additional blood sampling or intervention was performed. Medical records were reviewed to obtain demographic, clinical, and laboratory data. A total of 52 patients were treated with oxygen support, antibiotics, antiviral agents (Lopinavir/Ritonavir), or hydroxychloroquine.

The SOFA score, World Health Organization (WHO) disease severity, and 4C mortality score were assessed as described previously [3,5,6,31]. Critical disease included ARDS, sepsis, septic shock, and acute thrombosis [3,5,31]. 4C mortality score risk group was defined according to the scores; low (0–3), intermediate (4–8), high (9–14), and very high (≥15) [6]. For statistical analysis, three groups (low/intermediate, high, and very high) were analyzed due to the too small-sized low group. Clinical outcomes included ICU admission, ventilator use, ECMO use, and 30-day mortality [31].

### 2.2. sST2 Assay

A total of 52 residual EDTA plasma samples were collected from 52 patients at enrollment. The collected residual EDTA samples were aliquoted to avoid repeated freezing and thawing and were stored at −70 °C until measurement. Frozen samples were thawed at room temperature and gently mixed immediately before measuring sST2 levels. sST2 was assayed using the ichroma ST2 assay (Boditech Med Inc., Chuncheon-si, Gang-won-do, Republic of Korea) with an ichroma Ⅱ immuno-analyzer (Boditech Med Inc.) based on a fluorescence immunoassay. Both the ichroma ST2 assay and the ichroma Ⅱ immune-analyzer were approved by the Republic of Korean Ministry of Food and Drug Safety.

The manufacturer’s upper reference limit (URL) was 35 ng/mL, which was established as the cut-off value for predicting HF [13,15,18,21]. Analytical measurement intervals were from 3.1 to 200.0 ng/mL. sST2 levels were measured according to the manufacturer’s instructions.

### 2.3. Statistical Analysis

Data were presented as the number (percentage) or median (interquartile range, IQR). The Shapiro–Wilk test was used to determine the normality of data distribution. The Mann–Whitney U test was used to compare continuous variables (sST2, SOFA score, and 4C mortality score) according to the clinical outcomes. Chi-squared test or Fisher’s exact test was used to compare categorical variables (critical disease and 4C mortality risk group) according to the clinical outcomes. sST2, SOFA score, and critical disease were compared according to the 4C mortality risk group (low/intermediate, high, and very high) using the Kruskal–Wallis test or chi-squared test. 

With the receiver operating characteristic (ROC) curve analysis, the area under the curve (AUC), the optimal cut-off values, sensitivity, and specificity of sST2, SOFA score, critical disease, and 4C mortality were estimated to predict clinical outcomes [32]. A 95% confidence interval (CI) of AUC was calculated using the Mann–Whitney statistic approach, which was suggested as superior to others for the small sample size [32,33]. Kaplan–Meier survival analysis was used to estimate the hazard ratio (HR) with a 95% confidence interval (CI) for the 30-day mortality of sST2, SOFA score, critical disease, 4C mortality score, and 4C mortality score group. HRs (95% CI) of high and very high groups relative to low/intermediate groups were calculated.

The sample size for the Kaplan–Meier survival analysis was estimated based on the previous study [34]. The inputs were identical to those described in our previous study, except for the alternative survival probability; analysis time *t* = 1 month, accrual time α = 5 months, follow-up time *b* = 1-month, null survival probability, S_0_(*t*) = 0.013, 0.025, or 0.026, type I error rate (α) = 0.05, and the power (1 − β) = 0.8 [5]. The alternative survival probability was set to S_1_(*t*) = 0.231 based on the 30-day mortality of this study. Using log-minus-log transformation, which was suggested for improving the accuracy of the small sample size, the estimated sample size was between 11 and 15. Accordingly, the sample size of 52 was considered sufficient to perform the Kaplan–Meier survival analysis. MedCalc Software (version 20.111, MedCalc Software, Ostend, Belgium) was used for statistical analysis. *p*-value < 0.05 was considered statistically significant.

## 3. Results

The basic characteristics of the study population are summarized in Table 1. The median age (IQR) was 71.0 years (62.5–79.0), and males were 61.5% (*n* = 32). Among six patients with chronic cardiac disease, three patients had HF. The median sST2 level, SOFA score, and 4C mortality score were 46.6 ng/mL, 4.0, and 9.5, respectively. In WHO disease severity, critical disease was 71.2% (*n* = 37); sepsis and septic shock were 83.8% (*n* = 31) and 16.2% (*n* = 6), respectively. Eight sepsis and five septic shock patients had ARDS. Of a total of 52 patients, 26.9% patients admitted to ICU (*n* = 14). Among the 14 ICU patients, a ventilator was applied to 12 patients, and ECMO was applied to seven ventilated patients. The 30-day mortality was 23.1% (*n* = 12).

The sST2, SOFA score, critical disease, and 4C morality score according to clinical outcomes are presented in Table 2. The sST2 level and SOFA score differed significantly according to all clinical outcomes (all *p* < 0.05). The proportion of critical disease did not differ according to ventilator use and 30-day mortality, but the 4C mortality score differed significantly according to only the 30-day mortality. The sST2 level differed significantly among 4C mortality score risk groups (*p* = 0.042); however, the SOFA score and the proportion of critical disease did not (Table 3).

In the ROC curve analysis, the sST2 and SOFA score comparably predicted ICU admission, ventilator use, ECMO use, and 30-day mortality (sST2 vs. SOFA, all *p* > 0.05) (Figure 2). sST2 predicted ICU admission, ventilator use, ECMO use, and 30-day mortality better than critical disease (all *p* < 0.05). sST2 predicted ICU admission, ventilator use, and ECMO use better than the 4C mortality score (all *p* < 0.05). The sST2 and 4C mortality scores predicted comparably 30-day mortality (0.826 vs. 0.830, *p* = 0.963).

In the Kaplan–Meier survival analysis, the HR (95% CI) for predicting 30-day mortality was 8.4 (2.7–26.8) for sST2, 14.8 (3.0–71.7) for the SOFA score, 1.8 (0.5–6.5) for critical disease, and 11.7 (3.4–40.1) for the 4C mortality score (Figure 3). In 4C morality score risk groups, the 30-day survival probability differed significantly among low/intermediate, high, and very high groups (log-rank test, *p* < 0.001); HR (95% CI) was 4.3 (1.3–4.4) in the high group and 20.8 (1.9–221.2) in the very high group relative to the low/intermediate risk group (data not shown).

## 4. Discussion

This is the first study to explore the prognostic performance of the sST2, SOFA score, critical disease, and 4C mortality score simultaneously in hospitalized COVID-19 patients. In this study, most patients presented with critical disease. The median sST2 level of the total study population was 46.6 ng/mL, which was greater than the cut-off value for predicting HF, 35 ng/mL [13,15,18,21]. In this study, 25 of 29 patients with sST2 levels above 35 ng/mL had critical disease, and all 13 ARDS patients were included in these 25 patients. Among the 25 patients with critical disease, 20 patients presented dyspnea, which was known as the key symptom of HF [12]. It was not clear how many patients had HF because there was no further evaluation for the diagnosis of it. Thus, sST2 may have been elevated when reflecting HF, the degree of inflammation, or both. In the previous studies, the sST2 level was heterogeneous in COVID-19 [22,23,24,25,27,28,29,30]. In two previous studies, the overall sST2 levels were less than 10 ng/mL [22,23]. In other previous studies, median sST2 levels were from 48 ng/mL to 53.1 ng/mL in COVID-19 patients, similar to our data [24,28,29]. In this study, around 70% of patients had a 4C mortality score of nine or higher, belonging to the high and very high groups. Increasing age is the strongest variable in the 4C mortality score [6]. The age range from 70 to 79 is six points, and older than 80 is seven points. [6]. In this study, most of the patients were over the age of 70; therefore, most of them were given a score of six or higher. 

Both the sST2 level and SOFA score were significantly associated with ICU admission, ventilator use, ECMO use, and 30-day mortality. Median sST2 levels were lower than the URL in GW patients, patients without a ventilator and/or ECMO use, and survivors; however, it was about five- to ten-fold higher in ICU patients, patients with ventilator and/or ECMO use, and non-survivors. Based on our data, sST2 seems to reflect clinical outcomes in COVID-19. Similar to our data, a higher sST2 level was significantly associated with ICU admission and mortality [24,25,28,30]. The median sST2 level in non-survivors was 107 ng/mL, and it was higher than that of survived ICU and GW patients; the median sST2 level was significantly higher in ventilated ICU patients than in GW patients [28]. 

In COVID-19, the mechanism for the release of sST2 is not fully understood. IL-33 is released mainly by injured epithelial alveolar cells and can be upregulated in COVID-19 [18,35]. COVID-19 induces IL-33 expression in the lungs, and activated IL-33 leads to the production of sST2 type Ⅱ pneumocytes [8,35]. In COVID-19, ARDS, septic shock, or inflammatory mediators can lead to HF through multiple mechanisms [9], and HF can induce the upregulation of sST2 in the lungs leading to sST2 secretion by type Ⅱ pneumocytes [36]. 

Our data demonstrated that the sST2 level reflected a 4C mortality score better than the SOFA score and critical disease. Unlike the SOFA score and critical disease, the median sST2 level increased significantly according to the 4C mortality score risk group. In-hospital mortality ranged from 0.0% to 1.7% in the low-risk group and 8.0% to 9.9% in the intermediate-risk group, which were substantially lower than that of the high- and very high-risk groups [6,37]. In line with the previous studies, the median sST2 level was normal in the low/intermediate risk group, and it was significantly lower than that of the high-risk group. Based on our data, the sST2 level seems to reflect the prognostic index for COVID-19 mortality.

In this study, the sST2 level and SOFA score comparably predicted ICU admission, ventilator use, ECMO use, and 30-day mortality. AUCs of the sST2 level was higher than that of the SOFA score for predicting ICU admission and 30-day mortality (0.878 vs. 0.865 in ICU admission; 0.826 vs. 0.716 in 30-day mortality). sST2 outperformed critical disease in ICU admission, ventilator use, ECMO use, and 30-day mortality. Although the sST2 level and 4C mortality score comparably predicted 30-day mortality, sST2 outperformed the 4C mortality score for predicting ICU admission, ventilator use, and ECMO use. HRs for the sST2, SOFA score, and 4C mortality score were significantly high except for critical disease. Based on our data, sST2 might be a useful biomarker for predicting clinical outcomes in COVID-19. Since COVID-19 patients could be aggravated rapidly, resulting in unexpected poor outcomes, it is important to predict COVID-19 prognosis earlier [31]. The WHO recommends monitoring vital signs, clinical warning scores, laboratory data, electrocardiogram, or chest imaging to detect a deteriorating COVID-19 patient or complications [3,31]. These signs may not be monitored in a timely manner. In addition, clinical assessments such as the SOFA score, WHO disease severity, and 4C mortality score need various clinical and laboratory data or complex calculations. A simple biomarker would be more objective and appropriate in the real hospital setting [31]. Based on our data, sST2 could be used as a simple, objective parameter for detecting patients with critical care demands, including ICU, ventilator, and ECMO, and predicting 30-day mortality in clinical practice [31]. 

sST2 was superior to CRP and IL-6 for predicting ICU admission and mortality [24]. sST2 and clinical scores comparably predicted ICU admission for ventilator use and in-hospital mortality [29]. sST2 has a low biological variability and reference change values, and it is not affected by sex, age, body mass index, atrial fibrillation, renal function, or the prior diagnosis of HF [12]. In serial measurements, the sST2 level showed highly dynamic change according to the disease course in both COVID-19 and HF [23,24,25,36]. Accordingly, sST2 may be a reliable biomarker for the serial monitoring of hospitalized COVID-19 patients [12,38]. On the other hand, it is also necessary to consider the analytical performance and difference of sST2 assays to implement sST2 as a prognostic biomarker for COVID-19 [13,38,39,40]. 

This study had several limitations. First, although a sample size of 52 was sufficient for the Kaplan–Meier survival analysis, it might not be sufficient for providing a meaningful result. In addition, a smaller sample size would result in a wide CI because of the margin of error. However, a previous study reported that larger sample sizes reduce variance but do not improve AUC regression [41]. Further studies with a larger sample size are needed to validate our findings. Second, the study population had a skewed distribution toward critical disease, and our data may be biased and not representative. Third, the duration from the symptom onset to routine blood tests after admission varied, and it was difficult to obtain blood samples at a fixed time during the first and second waves of the COVID-19 pandemic [5,42,43]. The heterogeneous disease course of COVID-19 may have affected our data. Fourth, we focused on the prognostic utility of sST2 for clinical outcomes in COVID-19. The prediction of HF due to COVID-19 was out of the scope of our study due to insufficient information on it.

In conclusion, this is the first study to explore the prognostic utility of sST2 for predicting clinical outcomes in hospitalized COVID-19 patients compared with SOFA score, critical disease, and 4C mortality score. sST2 predicted ICU admission, ventilator use, ECMO use, and 30-day mortality comparably to SOFA score but significantly better than critical disease. sST2 predicted ICU admission, ventilator use, and ECMO use significantly better than the 4C mortality score. sST2 could be a useful biomarker for predicting clinical outcomes in hospitalized COVID-19 patients. Further studies are needed to implement sST2 as a prognostic biomarker for COVID-19 in routine clinical practice.

## Figures and Tables

**Figure 1 diagnostics-13-00259-f001:**
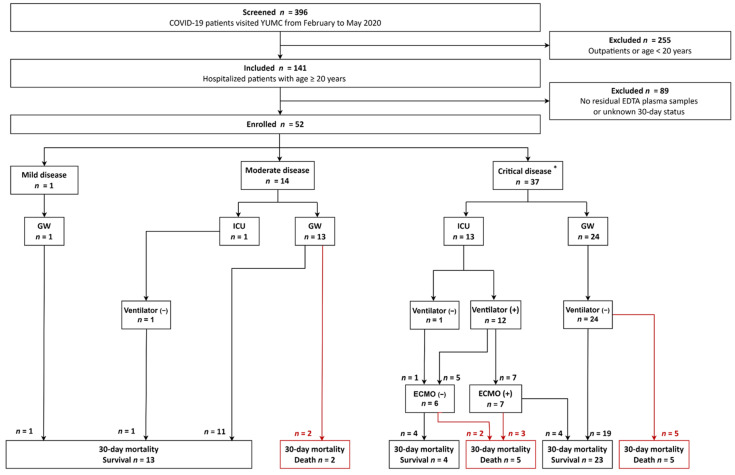
The enrollment and clinical outcomes of the study population. None of the study population presented severe disease in terms of WHO disease severity. The red line indicates death with regard to 30-day mortality. * Among 37 patients with critical disease, 31 patients had sepsis and six patients had septic shock. Abbreviations: WHO, World Health Organization; COVID-19, coronavirus disease 2019; YUMC, Yeoungnam University Medical Center; EDTA, ethylene-diamine-tetraacetic acid; ICU, intensive care unit; GW, general ward; ECMO, extracorporeal membrane oxygenation.

**Figure 2 diagnostics-13-00259-f002:**
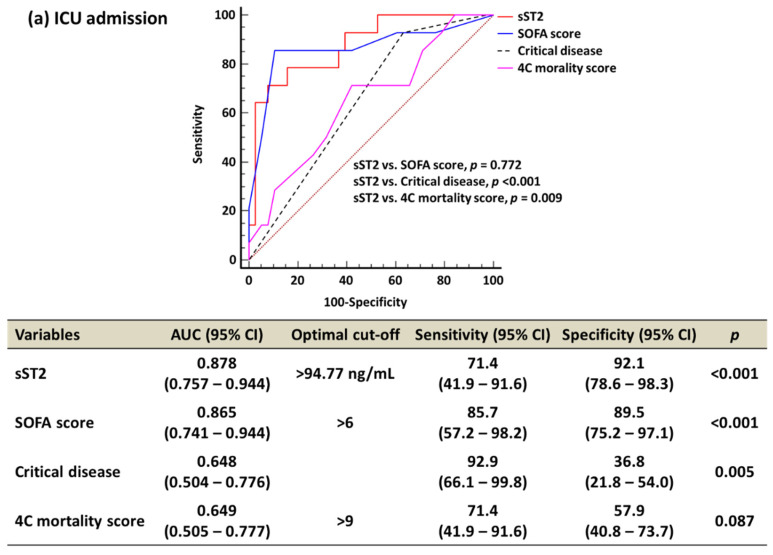
Receiver operating characteristic curve analysis of sST2, SOFA score, critical disease, and 4C mortality score for clinical outcomes. (**a**) ICU admission. (**b**) Ventilator use. (**c**) ECMO use. (**d**) 30-day mortality. Abbreviations: sST2—soluble suppressor of tumorigenicity 2; SOFA—sequential organ failure assessment; AUC—area under the curve; CI—confidence interval.

**Figure 3 diagnostics-13-00259-f003:**
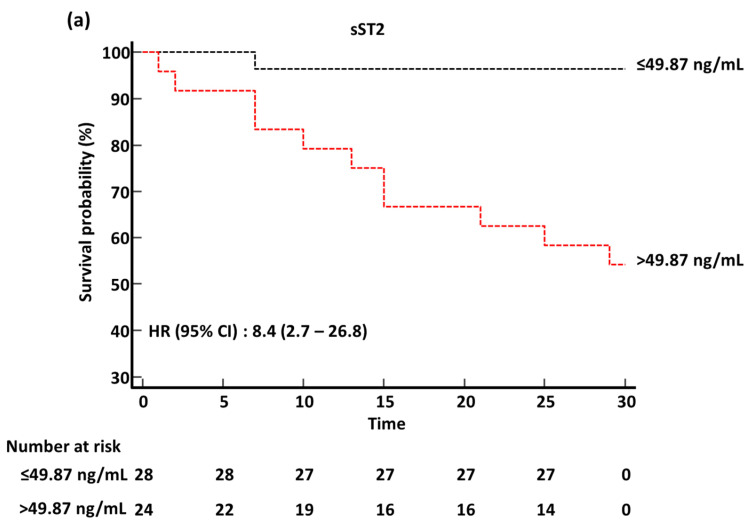
Kaplan–Meier survival analysis of sST2, SOFA score, critical disease, and 4C mortality score for 30-day mortality. (**a**) sST2. (**b**) SOFA score. (**c**) Critical disease. (**d**) 4C mortality score. Abbreviations: sST2, soluble suppressor of tumorigenicity 2; SOFA, sequential organ failure assessment; HR, hazard ratio; CI, confidence interval.

**Table 1 diagnostics-13-00259-t001:** The characteristics of the study population.

Variable	All Patients (*n* = 52)
Age, years	71.0 (62.5–79.0)
Male	32 (61.5)
BMI, kg/m^2^	24.1 (22.3–25.6)
Comorbidities	
DM	18 (34.6)
Malignancy	9 (17.3)
Chronic neurological conditions	7 (13.7)
Chronic cardiac disease *	6 (11.5)
Chronic respiratory disease (except asthma)	5 (9.6)
Dementia	5 (9.6)
Chronic kidney disease	3 (5.8)
Connective tissue disease	3 (5.8)
Others ^†^	3 (5.8)
Symptoms	
Fever or chilling	33 (63.5)
General weakness	23 (44.2)
Dyspnea	28 (53.8)
Cough	22 (42.3)
Sputum	14 (26.9)
Gastrointestinal ^‡^	7 (13.5)
Neurological ^∥^	6 (11.5)
Myalgia	5 (9.6)
Others ^¶^	5 (9.6)
Symptom duration, days	4.0 (0.5–7.5)
COVID-19 diagnosis to admission, days	0.0 (0.0–2.0)
COVID-19 diagnosis to blood sampling, days	3.0 (0.0–11.0)
Hospital stays, days	28.0 (16.5–47.5)
Treatment	
Oxygen support	42 (80.8)
Antibiotics	30 (57.7)
Antiviral agent (Lopinavir/Ritonavir)	25 (48.1)
Hydroxychloroquine	3 (5.8)
Severity assessment	
SOFA score	4.0 (1.0–7.0)
WHO disease severity	
Mild disease	1 (1.9)
Moderate disease	14 (26.9)
Severe disease	0 (0.0)
Critical disease	37 (71.2)
Sepsis/septic shock **	31 (83.8)/6 (16.2)
Prognosis assessment	
4C mortality score	9.5 (7.5–12.0)
Low (0–3)	2 (3.8)
Intermediate (4–8)	15 (28.8)
High (9–14)	30 (57.7)
Very high (≥15)	5 (9.6)
Clinical outcomes	
ICU admission	14 (26.9)
Ventilator use	12 (23.1)
ECMO use	7 (13.5)
30-day mortality	12 (23.1)
Vital signs	
SBP, mm Hg	120.0 (110.0–140.0)
DBP, mm Hg	71.5 (70.0–80.0)
HR, beats/min	82.0 (74.0–93.5)
RR, breaths/min	20.0 (20.0–23.0)
BT, ℃	37.1 (36.8–37.8)
SpO_2_, %	96 (94.3–97.0)
Laboratory data	
WBC, ×10^9^/L	5.9 (4.8–8.6)
Hb, g/L	120.0 (109.0–130.5)
PLT, ×10^9^/L	199.0 (148.0–268.0)
AST, U/L	30.5 (23.0–44.0)
ALT, U/L	24.5 (14.5–37.5)
Total bilirubin, umol/L	12.7 (9.6–18.3)
Cr, umol/L	76.5 (56.6–96.4)
Lactate, mmol/L	1.8 (1.2–2.2)
CRP, mg/L	36.4 (4.6–132.5)
NT-proBNP ^††^, pg/mL	230.2 (111.6–852.3)
hs-TnI ^‡‡^, ng/mL	0.01 (0.00–0.02)
sST2, ng/mL	46.6 (18.8–106.6)

Data are represented as a number (percentage) or median (interquartile range, IQR). * Chronic cardiac disease includes coronary artery disease (*n* = 3) and heart failure (*n* = 3). ^†^ Others include clinician-defined obesity (*n* = 2) and liver disease (*n* = 1). ^‡^ Gastrointestinal symptoms include anorexia, nausea, diarrhea, and abdominal distension/pain. ^∥^ Neurological symptoms include headache, dizziness, mental change, and motor weakness. ^¶^ Other symptoms include hemoptysis, rhinorrhea, chest pain, nasal congestion, and sore throat. ** 13 patients (sepsis [*n* = 8], septic shock [*n* = 5]) had ARDS. One sepsis patient without ARDS had acute thrombosis (acute stroke). ^††^ NT-proBNP levels were obtained from 38 patients at enrollment. ^‡‡^ hs-TnI levels were obtained from 37 patients at enrollment. Abbreviations: BMI, body mass index; DM, diabetes mellitus; COVID-19, coronavirus disease 2019; SOFA, sequential organ failure assessment; WHO, World Health Organization; ICU, intensive care unit; ECMO, extracorporeal membrane oxygenation; SBP, systolic blood pressure; DBP, diastolic blood pressure; HR, heart rate; RR, respiratory rate; BT, body temperature; SpO_2_, peripheral oxygen saturation; WBC, white blood cell; Hb, hemoglobin; PLT, platelets; AST, aspartate aminotransferase; ALT, alanine aminotransferase; Cr, creatinine; CRP, C-reactive protein; NT-proBNP, N-terminal pro-B-type natriuretic peptide; hs-TnI, high-sensitivity troponin I; sST2, soluble suppressor of tumorigenicity 2; ARDS, acute respiratory distress syndrome.

**Table 2 diagnostics-13-00259-t002:** sST2, SOFA score, critical disease, and 4C mortality score according to clinical outcomes.

Variables	ICU Admission			Ventilator Use			ECMO Use			30-Day Mortality		
	Yes (*n* = 14)	No (*n* = 38)	*p* *	Yes (*n* = 12)	No (*n* = 40)	*p* *	Yes (*n* = 7)	No (*n* = 45)	*p* *	Yes (*n* = 12)	No (*n* = 40)	*p* *
sST2, ng/mL	242.7 (77.8–378.0)	28.0 (16.3–59.5)	<0.001	295.1 (137.3–412.5)	28.1 (16.3–59.1)	<0.001	252.5 (146.9–667.9)	30.0 (17.6–76.9)	0.002	154.8 (59.1–426.6)	28.1 (16.3–79.2)	<0.001
SOFA score	7.5 (7.0–9.0)	3.0 (1.0–4.0)	<0.001	8.0 (7.0–9.5)	3.0 (0.5–4.0)	<0.001	8.0 (7.3–9.0)	3.0 (1.0–5.0)	<0.001	7.5 (3.0–9.5)	4.0 (0.5–5.5)	0.023
Critical disease	13 (92.9)	24 (63.2)	<0.043	12 (100.0)	25 (62.5)	0.011	7 (100.0)	30 (66.7)	0.093	10 (83.3)	27 (67.5)	0.470
4C mortality score	10.5 (8.0–14.0)	9.0 (7.0–12.0)	0.099	11.5 (9.0–14.0)	9.0 (7.0–11.5)	0.039	11.0 (8.5–12.8)	9.0 (7.0–12.0)	0.367	13.0 (11.5–15.5)	9.0 (7.0–10.0)	<0.001
Low/intermediate	4 (28.6)	13 (34.2)	0.765	3 (25.0)	14 (35.0)	0.579	2 (28.6)	15 (33.3)	0.582	1 (8.3)	16 (40.0)	0.002
High	8 (57.1)	22 (57.9)		7 (58.3)	23 (57.5)		5 (71.4)	25 (55.6)		7 (58.3)	23 (57.5)	
Very high	2 (14.3)	3 (7.9)		2 (16.7)	3 (7.5)		0 (0.0)	5 (11.1)		4 (33.3)	1 (2.5)	

Data were represented as a number (percentage) or median (IQR). * *p*-values were calculated using the Mann–Whitney test, chi-squared test, or Fisher’s exact test. Abbreviations: sST2, soluble suppressor of tumorigenicity 2; SOFA, sequential organ failure assessment; ICU, intensive care unit; ECMO, extracorporeal membrane oxygenation.

**Table 3 diagnostics-13-00259-t003:** sST2, SOFA score, and critical disease according to 4C mortality score risk group.

Variables	Low/Intermediate (*n* = 17)	High (*n* = 30)	Very High (*n* = 5)	*p* *
sST2, ng/mL	19.2 (12.9–58.7)	51.8 (22.9–118.3)	76.6 (49.5–347.8)	0.042 ^†^
SOFA score	3.0 (0.0–6.3)	4.0 (2.0–7.0)	8.0 (3.0–10.3)	0.093
Critical disease	9 (52.9)	23 (76.7)	5 (100.0)	0.073

Data were represented as a number (percentage) or median (IQR). * *p*-values were calculated using the Kruskal–Wallis test or chi-squared test among low/intermediate, high, and very high groups. ^†^ *p*-values between two groups calculated using Mann–Whitney test; Low/intermediate vs. High, *p* = 0.030; Low/intermediate vs. Very high, *p* = 0.055; High vs. Very high, *p* = 0.409. Abbreviations: sST2, soluble suppressor of tumorigenicity 2; SOFA, sequential organ failure assessment.

## Data Availability

The data presented in this study are available on request from the corresponding author.

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
