# Peer review of "Soluble ST2 as a Useful Biomarker for Predicting Clinical Outcomes in Hospitalized COVID-19 Patients"

_diagnostics, 2023, doi:10.3390/diagnostics13020259_

Round 1

Reviewer 1 Report

Park et al. have performed a prognostic biomarker study of sST2 in patients with COVID-19.

Major Comments

Sample size. A sample size of only 52 patients is likely insufficient to provide a meaningful result and as such should be considered hypothesis generating. Thus, please amend your limitations – 52 patients are insufficient. Additionally, there was no validation cohort. Has a statistician been consulted? Further evidence of this is the wide confidence interval margins for your AUROC plots.

Patient recruitment and details. Were patients recruited at the same point in their illness cycle? E.g. within 5 days of illness onset. Did the patients receive other treatments? What was the duration of follow up? Did patients have any specific limitations on care at enrolment (e.g. not for resuscitation orders).

Minor Comments

Study population in materials/methods – this section seems more detailed with regards to the disease severity etc. The description of the patients should be in results, but the recruitment methods should be discussed in the methods.

Table 1 – the left margin is cut off. Requires reformatting.

Table 2 – the right margin against the edge of the page. Requires reformatting.

The manuscript has multiple minor grammatical errors and requires further editing.

Author Response

Response to Reviewer 1 Comments

Point 1: Park et al. have performed a prognostic biomarker study of sST2 in patients with COVID-19.

Major Comments

Sample size. A sample size of only 52 patients is likely insufficient to provide a meaningful result and as such should be considered hypothesis generating. Thus, please amend your limitations – 52 patients are insufficient. Additionally, there was no validation cohort. Has a statistician been consulted? Further evidence of this is the wide confidence interval margins for your AUROC plots.

Response 1: Our co-author Prof. Seungho Lee provided statistical consulting. The sample size was reduced by considering the inclusion criteria in our study. Hence, we performed the non-parametric statistical approach for ROC curve analysis. Because of the margin of error, a smaller sample size would result a wide confidence interval. The previous reported that a large sample size reduces variance, but the AUC regression does not improve. Thus, we provided our results with the small sample size. Like your comment, we will conduct a validation study with the larger study subjects in the near future. We revised the sentences in statistical analysis and discussion. We added new references on statistical methods. We highlighted the changed portions by using red-colored text. We hope that you will find our paper suitable for publication.

2.3. Statistical analysis

  Data were presented as number (percentage) or median (interquartile range, IQR). The Shapiro–Wilk test was used to determine the normality of data distribution. The Mann–Whitney U test was used to compare continuous variables (sST2, SOFA score, and 4C mortality score) according to clinical outcomes. Chi-squared test or Fisher’s exact test were used to compare categorical variables (critical disease and 4C mortality risk group) according to clinical outcomes. sST2, SOFA score, and critical disease were compared according to 4C mortality risk group (low/intermediate, high, and very high) using the Kruskal-Wallis test or chi-squared test.

  With the receiver operating characteristic (ROC) curve analysis, area under the curve (AUC), the optimal cut-off values, sensitivity, and specificity of sST2, SOFA score, critical disease, and 4C mortality were estimated to predict clinical outcomes [32]. 95% confidence interval (CI) of AUC was calculated using the Mann-Whitney statistic approach, which was suggested to be superior to others for small sample size [32, 33]. Kaplan–Meier survival analysis was used to estimate the hazard ratio (HR) with 95% confidence interval (CI) for 30-day mortality of sST2, SOFA score, critical disease, 4C mortality score, and 4C mortality score group. HRs (95% CI) of high and very high groups relative to low/intermediate group were calculated.

  The sample size for the Kaplan-Meier survival analysis was estimated based on the previous study [34]. The inputs were identical to those described in our previous study, except for the alternative survival probability; analysis time t = 1 month, accrual time α = 5 months, follow-up time b =1 month, null survival probability, S0(t) = 0.013, 0.025, or 0.026, typeⅠerror rate (α) = 0.05, and the power (1 – β) = 0.8 [5]. The alternative survival probability was set to set to S1(t) = 0.231 based on 30-day mortality of this study. Using log-minus-log transformation suggested for improving accuracy in small sample size, the estimated sample size was 11 to 15. Accordingly, the sample size of 52 was considered sufficient to perform the Kaplan-Meier survival analysis. MedCalc Software (version 20.111, MedCalc Software, Ostend, Belgium) was used for statistical analysis. p value < 0.05 was considered statistically significant. (Pages 3-4)

4. Discussion

  This study had several limitations. First, although a sample size of 52 was sufficient for the Kaplan-Meier survival analysis, it may not be sufficient to provide a meaningful result. Besides, a smaller sample size would result a wide CI because of the margin of error. However, a previous study reported that larger sample size reduces variance but does not improve AUC regression [41]. Further studies with the larger sample size are needed to validate our findings. Second, the study population had a skewed distribution towards critical disease, and our data may be biased and not be representative. Third, the duration from symptom onset to routine blood tests after admission varied, and it was difficult to obtain blood samples at a fixed time during the first and second wave of the COVID-19 pandemic [5, 42-43]. The heterogeneous disease course of COVID-19 may have affected our data. Fourth, we focused the prognostic utility of sST2 for clinical outcomes in COVID-19. The prediction of HF due to COVID-19 was out of the scope of our study due to insufficient information on it. (Pages 13-14)

<Added references>

32. DeLong, E. R.; DeLong, D. M.; Clarke-Pearson, D. L. Comparing the Areas under Two or More Correlated Receiver Operating Characteristic Curves: A Nonparametric Approach. Biometrics 1988, 44, 837-845.

33. Feng, D.; Cortese, G.; Baumgartner, R. A Comparison of Confidence/Credible Interval Methods for the Area Under the ROC Curve for Continuous Diagnostic Tests With Small Sample Size. Stat Methods Med Res 2017, 26, 2603-2621.

34. Nagashima, K.; Noma, H.; Sato, Y.; Gosho, M. Sample Size Calculations for Single‐Arm Survival Studies Using Transformations of the Kaplan–Meier Estimator. Pharm Stat 2020, 20, 499–511.

41. Hanczar, B.; Hua, J.; Sima, C.; Weinstein, J.; Bittner, M.; Dougherty, E. R. Small-sample precision of ROC-related estimates. Bioinformatics 2010, 26, 822-830.

Point 2: Patient recruitment and details. Were patients recruited at the same point in their illness cycle? E.g. within 5 days of illness onset. Did the patients receive other treatments? What was the duration of follow up? Did patients have any specific limitations on care at enrolment (e.g. not for resuscitation orders).

Response 2: We revised the sentences in study population. We highlighted the changed portions by using red-colored text. In addition, we added treatment in Table 1. See Table 1 on pages 4-6.

2. Materials and Methods

2.1. Study population

  The enrollment and clinical outcomes of the study population are presented in Figure. 1. From February to May 2020, a total of 396 COVID-19 patients Yeungnam University Medical Center (YUMC), Daegu, Korea. We excluded 255 patients visited the outpatient clinic without routine blood tests in clinical practice or were younger than 20 years. From 141 hospitalized patients with over 20 years of age, we further excluded 89 patients without available residual ethylene-diamine-tetraacetic acid (EDTA) plasma samples after routine blood tests and known 30-day status. Finally, 52 patients were enrolled. Among 52 patients, 44 patients were included from a previous study [5]. The study population had no specific limitations on care at enrollment. The symptom duration ranged up to 25 days. Therefore, we could not enroll the study population at the same point in the COVID-19 disease course. In a total of 52 patients, follow-up period for 30-day status was 5 months from February to the end of June 2020. No additional blood sampling or intervention was performed. Medical record was reviewed to obtain demographic, clinical, and laboratory data. 52 patients were treated by oxygen support, antibiotics, antiviral agent (Lopinavir/Ritonavir), or hydroxychroloquine.

  SOFA score, World Health Organization (WHO) disease severity, and 4C mortality score were assessed as described previously [3, 5, 6, 31]. Critical disease included ARDS, sepsis, septic shock, and acute thrombosis [3, 5, 31]. 4C mortality score risk group was defined according to scores; low (0 – 3), intermediate (4 – 8), high (9 – 14), and very high (≥ 15) [6]. For statistical analysis, three groups (low/intermediate, high, and very high) were analyzed due to too small-sized low group. Clinical outcomes included ICU admission, ventilator use, ECMO use, and 30-day mortality [31]. (Page 2-3)

Point 3: Minor Comments

Study population in materials/methods – this section seems more detailed with regards to the disease severity etc. The description of the patients should be in results, but the recruitment methods should be discussed in the methods.

Response 3: We highlighted the changed portions by using red-colored text. In addition, we revised Table 1. See Table 1 on pages 4-6.

3. Results

  Basic characteristics of the study population are summarized in Table 1. The median age (IQR) was 71.0 years (62.5 - 79.0), and males were 61.5% (n = 32). Among six patients with chronic cardiac disease, three patients had HF. Median sST2 level, SOFA score, and 4C mortality score were 46.6 ng/mL, 4.0, and 9.5, respectively. In WHO disease severity, critical disease was 71.2% (n = 37); sepsis and septic shock were 83.8% (n = 31) and 16.2% (n = 6), respectively. Eight sepsis and five septic shock patients had ARDS. In a total of 52 patients, 26.9% patients admitted to ICU (n = 14). Among the 14 ICU patients, ventilator was applied to 12 patients, and ECMO was applied to seven ventilated patients. 30-day mortality was 23.1% (n = 12). (Page 4)

Point 4: Table 1 – the left margin is cut off. Requires reformatting.

Response 4: We revised Table 1 and highlighted the changed portions by using red-colored text. Please see Table 1 on pages 4-6.

Point 5: Table 2 – the right margin against the edge of the page. Requires reformatting.

Response 5: We revised Table 2 and highlighted the changed portions by using red-colored text. See Table 2 on page 7.

Point 6: The manuscript has multiple minor grammatical errors and requires further editing.

Response 6: We revised grammatical errors throughout the manuscript. We highlighted the changed portions by using red-colored text. 

Reviewer 2 Report

a) in the introduction, include a paragraph describing the sections of the article

b) mathematic and methods must be expanded. justify the choice of the statistical method. explain in more detail the statistical method used.

c) the number of citations on statistical methods should be increased

Author Response

Response to Reviewer 2 Comments

Point 1: a) in the introduction, include a paragraph describing the sections of the article

Response 1: We revised the introduction. In adddition, we added new references. We highlighted the changed portions by using red-colored text. We hope that you will find our paper suitable for publication.

1. Introduction

  The coronavirus disease 2019 (COVID-19) pandemic has caused globally over six million deaths as of November 8, 2022, since its outbreak occurred in December 2019 [1, 2]. The clinical spectrum of COVID-19 ranges from asymptomatic to critical disease [3]. Critical disease can present acute respiratory distress syndrome (ARDS), sepsis, septic shock, thromboembolism, or multi-organ failure (MOF) [3-5]. MOF can occur in lung, heart, liver, kidney, and nervous system, and may also present with hematological abnormalities [4]. The sequential organ failure assessment (SOFA) sore is used as a severity index for assessing sepsis in COVID-19 [3, 5]. Recently, 4C mortality score was developed as a prognostic index to predict in-hospital mortality in COVID-19 [6]. 4C mortality score is calculated using age, sex, comorbidities, respiratory rate, peripheral oxygen saturation on room air, Glasgow Coma Scale, and levels urea and C-reactive protein (CRP), and it predicted in-hospital mortality better than SOFA score [6].

  When severe acute respiratory syndrome coronavirus 2 (SARS-CoV-2) enters host cells through angiotensin-converting enzyme 2 (ACE2) receptor, it releases inflammatory cytokines including interleukin (IL)-1, IL-6, or tissue necrosis factor-alpha (TNF-α) [4, 5, 7, 8]. Overproduced cytokines induce the cytokine storm leading to MOF or death [4, 7, 8]. COVID-19 can cause cardiovascular disease (CVD) including heart failure (HF), and it can lead to sudden cardiac death [3-4, 9-10]. COVID-19 patients with HF have greater risk of poor outcomes in COVID-19 [3, 9-11]. In COVID-19, MOF, ICU admission, and in-hospital mortality were significantly higher in the HF group than the non-HF group [11]. Cardiac biomarker could play an important role to predict the prognosis of COVID-19 [12]. Traditional biomarkers like natriuretic peptides and troponins were associated with COVID-19 disease severity and mortality [10, 12-13].

  Soluble suppression of tumorigenesis-2 (sST2), the soluble isoform of ST2 which is a member of the IL-1 superfamily, is an emerging prognostic biomarker for sepsis as well as for HF [12-17]. sST2 is not only released by vascular congestion but also by inflammatory and pro-fibrotic stimuli [14]. Previous studies suggested that sST2 might be a prognostic biomarker to predict the 30-day and in-hospital mortality in sepsis [14-17]. When IL-33 binds to sST2, IL-33/sST2 inhibits the release of anti-inflammatory cytokines, followed by the activation and the release of inflammatory cytokines like IL-6 and TNF-α leading to inflammation [13-18], In addition, the previous studies have been reported that sST2 would predict COVID-19 disease severity and prognosis [18-30]. sST2 level increased according to COVID-19 disease severity [23, 25]. sST2 was associated with intensive care unit (ICU) admission, ventilator use, thrombosis, and mortality in COVID-19 [22, 24-30].

  Our previous studies showed that biomarker-based approach would be useful to predict clinical outcomes in COVID-19 compared with clinical index including SOFA score and COVID-19 disease severity (critical disease) [5, 31]. To the best of our knowledge, few studies have evaluated the prognostic performance of sST2 compared with clinical assessment in COVID-19 [29]. We aimed to explore the prognostic utility of sST2 for predicting clinical outcomes in hospitalized COVID-19 patients compared with clinical assessment such as SOFA score, critical disease, and 4C mortality score. In addition, we explored sST2 levels according to 4C mortality score. (Pages 1-2)

<Added references>

14. Yang, H. S.; Kim, H. J.; Shim, H. J.; Kim, S. J.; Hur, M.; Di Somma, S. Soluble ST2 and Troponin I Combination: Useful Biomarker for Predicting Development of Stress Cardiomyopathy in Patients Admitted to the Medical Intensive Care Unit. Heart Lung 2015, 44, 282-288.

16. Kim, H.; Hur, M.; Moon, H.-W.; Yun, Y.-M.; Di Somma, S. Multi-Marker Approach Using Procalcitonin, Presepsin, Galectin-3, and Soluble Suppression of Tumorigenicity 2 for the Prediction of Mortality in Sepsis. Ann Intensive Care 2017, 7, 27.

Point 2: b) mathematic and methods must be expanded. justify the choice of the statistical method. explain in more detail the statistical method used.

Response 2: We highlighted the changed portions by using red-colored text.

2.3. Statistical analysis

  Data were presented as number (percentage) or median (interquartile range, IQR). The Shapiro–Wilk test was used to determine the normality of data distribution. The Mann–Whitney U test was used to compare continuous variables (sST2, SOFA score, and 4C mortality score) according to clinical outcomes. Chi-squared test or Fisher’s exact test were used to compare categorical variables (critical disease and 4C mortality risk group) according to clinical outcomes. sST2, SOFA score, and critical disease were compared according to 4C mortality risk group (low/intermediate, high, and very high) using the Kruskal-Wallis test or chi-squared test.

  With the receiver operating characteristic (ROC) curve analysis, area under the curve (AUC), the optimal cut-off values, sensitivity, and specificity of sST2, SOFA score, critical disease, and 4C mortality were estimated to predict clinical outcomes [32]. 95% confidence interval (CI) of AUC was calculated using the Mann-Whitney statistic approach, which was suggested to be superior to others for small sample size [32, 33]. Kaplan–Meier survival analysis was used to estimate the hazard ratio (HR) with 95% confidence interval (CI) for 30-day mortality of sST2, SOFA score, critical disease, 4C mortality score, and 4C mortality score group. HRs (95% CI) of high and very high groups relative to low/intermediate group were calculated.

  The sample size for the Kaplan-Meier survival analysis was estimated based on the previous study [34]. The inputs were identical to those described in our previous study, except for the alternative survival probability; analysis time t = 1 month, accrual time α = 5 months, follow-up time b =1 month, null survival probability, S0(t) = 0.013, 0.025, or 0.026, typeⅠerror rate (α) = 0.05, and the power (1 – β) = 0.8 [5]. The alternative survival probability was set to set to S1(t) = 0.231 based on 30-day mortality of this study. Using log-minus-log transformation suggested for improving accuracy in small sample size, the estimated sample size was 11 to 15. Accordingly, the sample size of 52 was considered sufficient to perform the Kaplan-Meier survival analysis. MedCalc Software (version 20.111, MedCalc Software, Ostend, Belgium) was used for statistical analysis. p value < 0.05 was considered statistically significant. (Pages 3-4)

Point 3: c) the number of citations on statistical methods should be increased

Response 3: We added new references on statistical methods.

<Added references>

32. DeLong, E. R.; DeLong, D. M.; Clarke-Pearson, D. L. Comparing the Areas under Two or More Correlated Receiver Operating Characteristic Curves: A Nonparametric Approach. Biometrics 1988, 44, 837-845.

33. Feng, D.; Cortese, G.; Baumgartner, R. A Comparison of Confidence/Credible Interval Methods for the Area Under the ROC Curve for Continuous Diagnostic Tests With Small Sample Size. Stat Methods Med Res 2017, 26, 2603-2621.

34. Nagashima, K.; Noma, H.; Sato, Y.; Gosho, M. Sample Size Calculations for Single‐Arm Survival Studies Using Transformations of the Kaplan–Meier Estimator. Pharm Stat 2020, 20, 499–511.

41. Hanczar, B.; Hua, J.; Sima, C.; Weinstein, J.; Bittner, M.; Dougherty, E. R. Small-sample precision of ROC-related estimates. Bioinformatics 2010, 26, 822-830.

Reviewer 3 Report

The ongoing COVID-19 pandemic caused more than 6 million of deaths. Thus, the scientists are looking for some cellular biomarkers which may estimate the course of SARS-CoV-2 infection. Park and colleagues showed that the soluble suppression of tumorigenesis-2 (sST-2) can be used as a biomarker to predict the poor prognosis of COVID-19. The paper is well-written and the results are quite clear. I have only some minor comments, which you can find below.

1.       Please change the title! Currently, it is the title of Special Issue.

2.       In the abstract, I would recommend to delete the subheadings, such as ‘Backgrounds’, ‘Methods’, ‘Results’ and ‘Conclusions’.

3.       In the Material and Methods, Study population, authors wrote that they excluded 396 out of 396 COVID-19 patients. I am afraid that in such case, there will be no one for analysis of anything.

4.       Please put the Figures in the manuscript just after they are mentioned in the text. Please read the Instructions for Authors.

5.       Finally, I would like to ask how specific can be the sST-22? Did authors consider that all critical COVID-19 patients has some symptoms of heart failure and therefore the sST-2 could be elevated?

Author Response

Response to Reviewer 3 Comments

The ongoing COVID-19 pandemic caused more than 6 million of deaths. Thus, the scientists are looking for some cellular biomarkers which may estimate the course of SARS-CoV-2 infection. Park and colleagues showed that the soluble suppression of tumorigenesis-2 (sST-2) can be used as a biomarker to predict the poor prognosis of COVID-19. The paper is well-written and the results are quite clear. I have only some minor comments, which you can find below.

Point 1: 1. Please change the title! Currently, it is the title of Special Issue.

Response 1: Thank you for your comment. We changed the title. We highlighted the changed portions by using red-colored text.

Article

Soluble ST2 as a Useful Biomarker for Predicting Clinical Outcomes in Hospitalized COVID-19 Patients

Point 2: 2. In the abstract, I would recommend to delete the subheadings, such as ‘Backgrounds’, ‘Methods’, ‘Results’ and ‘Conclusions’.

Response 2: We deleted the subheadings in the abstract. We highlighted the changed portions by using red-colored text.

Abstract: Soluble suppression of tumorigenesis-2 (sST2) is an emerging biomarker for sepsis as well as for heart failure. We investigated the prognostic utility of sST2 for predicting clinical outcomes in hospitalized coronavirus disease 2019 (COVID-19) patients. In a total of 52 hospitalized COVID-19 patients, sST2 levels were measured using the ichroma ST2 assay (Boditech Med Inc., Chuncheon-si, Gang-won-do, Korea). Clinical outcomes included intensive care unit (ICU) admission, ventilator use, extracorporeal membrane oxygenation (ECMO) use, and 30-day mortality. sST2 was analyzed according to clinical outcomes. sST2, sequential organ failure assessment (SOFA) score, critical disease, and 4C mortality score were compared using the receiver operating characteristic (ROC) curve and Kaplan-Meier methods for clinical outcomes. sST2 level differed significantly according to ICU admission, ventilator use, ECMO use, 30-day mortality (all p <0.05). On ROC curve analysis, sST2 predicted ICU admission, ventilator use, ECMO use, and 30-day mortality comparable to SOFA score, but significantly better than critical disease. sST2 predicted ICU admission, ventilator use, and ECMO use significantly better than 4C mortality score. On Kaplan -Meier survival analysis, hazard ratios (95% confidence interval) were 8.4 (2.7 – 26.8) for sST2, 14.8 (3.0 – 71.7) for SOFA score, 1.8 (0.5 – 6.5) for critical disease, and 11.7 (3.4 – 40.1) for 4C mortality score. This study demonstrated that sST2 could be a useful biomarker to predict ICU admission, ventilator use, ECMO use, and 30-day mortality in hospitalized COVID-19 patients. sST2 may be implemented as a prognostic COVID-19 biomarker in clinical practice. (Page 1)

Point 3: 3. In the Material and Methods, Study population, authors wrote that they excluded 396 out of 396 COVID-19 patients. I am afraid that in such case, there will be no one for analysis of anything.

Response 3: We corrected a typo error. We highlighted the changed portions by using red-colored text.

2. Materials and Methods

2.1. Study population

  The enrollment and clinical outcomes of the study population are presented in Figure. 1. From February to May 2020, a total of 396 COVID-19 patients Yeungnam University Medical Center (YUMC), Daegu, Korea. We excluded 255 patients visited the outpatient clinic without routine blood tests in clinical practice or were younger than 20 years. From 141 hospitalized patients with over 20 years of age, we further excluded 89 patients without available residual ethylene-diamine-tetraacetic acid (EDTA) plasma samples after routine blood tests and known 30-day status. Finally, 52 patients were enrolled. Among 52 patients, 44 patients were included from a previous study [5]. (Page 2)

Point 4: 4. Please put the Figures in the manuscript just after they are mentioned in the text. Please read the Instructions for Authors.

Response 4: We put the figures in the manuscript just after they are mentioned in the text according to the Instructions for Authors. See figures on pages 3, 8-12.

Point 5: 5. Finally, I would like to ask how specific can be the sST-22? Did authors consider that all critical COVID-19 patients has some symptoms of heart failure and therefore the sST-2 could be elevated?

Response 5: We highlighted the changed portions by using red-colored text.

4. Discussion

  This is the first study that explored the prognostic performance of sST2, SOFA score, critical disease, and 4C mortality score simultaneously in hospitalized COVID-19 patients. In this study, most patients presented critical disease. Median sST2 level of total study population was 46.6 ng/mL greater than the cut-off value for predicting HF, 35 ng/mL [13, 15, 18, 21]. In this study, 25 of 29 patients with sST2 levels above 35 ng/mL had critical disease, and all 13 ARDS patients were included in these 25 patients. Among the 25 patients with critical disease, 20 patients presented dyspnea, which was known as the key symptom of HF [12]. It was not clear how many patients had HF because there was no further evaluation for the diagnosis of it. Thus, sST2 may have been elevated reflecting HF, the degree of inflammation, or both. In the previous studies, sST2 level was heterogeneous in COVID-19 [22-25, 27-30]. In two previous studies, overall sST2 levels were less than 10 ng/mL [22-23]. In other previous studies, median sST2 levels were from 48 ng/mL to 53.1 ng/mL in COVID-19 patients similar to our data [24, 28, 29]. In this study, around 70% of patients had 4C mortality score of 9 or higher, belonging to the high and very high groups. Increasing age is the strongest variable in 4C mortality score [6]. The age of 70 to 79 is 6 points, and older than 80 is 7 points. [6]. In this study, most of patients were over the age of 70; therefore, most of them were given a score of six or higher. (Page 12)

4. Discussion

  This study had several limitations. First, although a sample size of 52 was sufficient for the Kaplan-Meier survival analysis, it may not be sufficient to provide a meaningful result. Besides, a smaller sample size would result a wide CI because of the margin of error. However, a previous study reported that larger sample size reduces variance but does not improve AUC regression [41]. Further studies with the larger sample size are needed to validate our findings. Second, the study population had a skewed distribution towards critical disease, and our data may be biased and not be representative. Third, the duration from symptom onset to routine blood tests after admission varied, and it was difficult to obtain blood samples at a fixed time during the first and second wave of the COVID-19 pandemic [5, 42-43]. The heterogeneous disease course of COVID-19 may have affected our data. Fourth, we focused the prognostic utility of sST2 for clinical outcomes in COVID-19. The prediction of HF due to COVID-19 was out of the scope of our study due to insufficient information on it. (Pages 13-14)

Round 2

Reviewer 1 Report

Thank you for your revisions. 

Reviewer 2 Report

bibliographic referencing can be improved, with the inclusion of more recent articles: I suggest including: ELLIS MOREIRA, MIGUEL ÂNGELO ; SIMÕES GOMES, CARLOS FRANCISCO ; DOS SANTOS, MARCOS; DA SILVA JÚNIOR, ANTONIO CARLOS ; DE ARAÚJO COSTA, IGOR PINHEIRO . Sensitivity Analysis by the PROMETHEE-GAIA method: Algorithms evaluation for COVID-19 prediction. PROCEDUA COMPUTER SCIENCE, v. 199, p. 431-438, 2022. 10.1016/j.procs.2022.01.052 BARBOSA DE PAULA, NATÁLIA OLIVEIRA ; DE ARAÚJO COSTA, IGOR PINHEIRO ; DRUMOND, PAULA; LELLIS MOREIRA, MIGUEL ÂNGELO ; SIMÕES GOMES, CARLOS FRANCISCO ; DOS SANTOS, MARCOS; DO NASCIMENTO MAÊDA, SERGIO MITIHIRO . Strategic support for the distribution of vaccines against Covid-19 to Brazilian remote areas: A multicriteria approach in the light of the ELECTRE-MOr method. PROCEDUA COMPUTER SCIENCE, v. 199, p. 40-47, 2022.doi.org/10.1016/j.procs.2022.01.006 COSTA, I. P. A. ; SANSEVERINO, A.M.; BARCELOS, M. R. S. ; BELDERRAIN, M. C. N. ; GOMES, C. F. S. ; SANTOS, M. Choosing flying hospitals in the fight against the COVID-19 pandemic: structuring and modeling a complex problem using the VFT and ELECTRE-MOr methods. IEEE Latin America Transactions, v. 19, p. 1099-1106, 2021. 10.1109/TLA.2021.9451257 COSTA, IGOR PINHEIRO DE ARAÚJO ; GOMES, C.F.S. ; TEIXEIRA, L. F. H. S. B. ; SANTOS, M. dos ; Sergio Maeda . Choosing a hospital assistance ship to fight the covid-19 pandemic. JOURNAL OF PUBLIC HEALTH (ONLINE), v. 54, p. 1-8, 2020. doi.org/10.11606/s1518-8787.2020054002792